# Multilayer Perceptron-Based Real-Time Intradialytic Hypotension Prediction Using Patient Baseline Information and Heart-Rate Variation

**DOI:** 10.3390/ijerph191610373

**Published:** 2022-08-20

**Authors:** Tae Wuk Bae, Min Seong Kim, Jong Won Park, Kee Koo Kwon, Kyu Hyung Kim

**Affiliations:** 1Daegu-Gyeongbuk Research Center, Electronics and Telecommunications Research Institute, Daegu 42994, Korea; 2Division of Nephrology, Department of Internal Medicine, College of Medicine, Yeungnam University, Daegu 42415, Korea

**Keywords:** intradialytic hypotension, multilayer perceptron, heart-rate, hemodialysis, real-time

## Abstract

Intradialytic hypotension (IDH) is a common side effect that occurs during hemodialysis and poses a great risk for dialysis patients. Many studies have been conducted so far to predict IDH, but most of these could not be applied in real-time because they used only underlying patient information or static patient disease information. In this study, we propose a multilayer perceptron (MP)-based IDH prediction model using heart rate (HR) information corresponding to time-series information and static data of patients. This study aimed to validate whether HR differences and HR slope information affect real-time IDH prediction in patients undergoing hemodialysis. Clinical data were collected from 80 hemodialysis patients from 9 September to 17 October 2020, in the artificial kidney room at Yeungnam University Medical Center (YUMC), Daegu, South Korea. The patients typically underwent hemodialysis 12 times during this period, 1 to 2 h per session. Therefore, the HR difference and HR slope information within up to 1 h before IDH occurrence were used as time-series input data for the MP model. Among the MP models using the number and data length of different hidden layers, the model using 60 min of data before the occurrence of two layers and IDH showed maximum performance, with an accuracy of 81.5%, a true positive rate of 73.8%, and positive predictive value of 87.3%. This study aimed to predict IDH in real-time by continuously supplying HR information to MP models along with static data such as age, diabetes, hypertension, and ultrafiltration. The current MP model was implemented using relatively limited parameters; however, its performance may be further improved by adding additional parameters in the future, further enabling real-time IDH prediction to play a supporting role for medical staff.

## 1. Introduction

Intradialytic hypotension (IDH), in which blood pressure (BP) drops rapidly during hemodialysis, is a major risk factor in dialysis patients. Previous studies have reported that IDH occurs in approximately 15–20% of patients during dialysis [1]. IDH indicates a decrease of more than 10 mmHg in mean arterial pressure or more than 20 mmHg in systolic blood pressure (SBP) [2,3]. IDH can pose various risks to the central nervous system, heart, and kidneys [4,5]. The frequent occurrence of IDH may be an important risk factor for increased mortality in dialysis patients [6,7]. It has also been widely reported that IDH can significantly impact sudden cardiac death by causing severe arrhythmia [8]. IDH is caused by the interaction between the ultrafiltration rate (UFR), arteriolar tone, and cardiac output (CO). IDH is mainly caused by a decrease in blood volume due to the withdrawal of fluid from the vascular compartment and insufficient refilling of fluid from the interstitial compartment into the vascular compartment during ultrafiltration (UF) [9]. In patients with end-stage renal disease, an imbalance between a decrease in central blood volume and an appropriate hemodynamic response can lead to impaired cardiac function and an impaired autonomic nervous system (ANS) [10]. During hemodialysis, the UF procedure removes fluid from the vascular compartment and displaces it from the interstitial compartment. Therefore, the UFR affects the rate of plasma replenishment. If the UFR exceeds the plasma recharge rate, then the IDH potential increases. Excessive UF volume can cause decreased CO, especially when compensatory mechanisms such as myocardial contraction and heart rate (HR) are not optimal [5]. Other causes may include impaired ANS function, peripheral vasoconstriction, and cardiovascular diseases such as atherosclerosis and left ventricular hypertrophy [11]. Additionally, autonomic neuropathy, diabetes, and antihypertensive medications increase the likelihood of IDH [12].

In the event of intravascular hypovolemia, various compensatory mechanisms are activated, such as cardiac responses to maintain CO and venous circulation, arteriolar vasoconstriction to increase the total peripheral resistance, and plasma recharging from interstitial and intracellular compartments [13,14]. ANS dysfunction also significantly influences the occurrence of IDH. In patients with normal autonomic function, increase in HR and systemic resistance were observed, whereas in patients with autonomic neuropathy, the overall systemic resistance decreased during IDH despite a fixed HR [15]. ANS dysfunction causes IDH, which leads to an inappropriate sympathetic response to hypovolemia that occurs during hemodialysis [16].

A common treatment for IDH is to keep the patient’s feet above the head and significantly slow down the UFR to slow the loss of blood volume due to fluid removal [9]. Another method is to inject a hypertonic solution that increases blood volume and BP [17]. As these practical measures are taken at the onset of symptoms in patients, it is very important to predict IDH in advance. Engineering approaches for predicting IDH have been adopted. It was assumed that the increased variability in short-term oxygen saturation resulted in changes in CO before IDH, which was used as a predictor of IDH [18]. Another approach assumed that peripheral vasoconstriction precedes IDH [19]. This method provided a warning when the size of the normalized photoplethysmogram (PPG) envelope was less than the threshold. Additionally, the HR turbulence induced by premature ventricular beats was studied for IDH prediction [20]. Abnormal HR turbulence refers to autonomic dysfunction, and autonomic neuropathy is closely associated with IDH [21]. Meanwhile, spectral power using heart-rate variability (HRV) obtained from the measured electrocardiogram (ECG) was studied to identify patients predisposed to IDH [20,22]. For example, in episodes without hypotension, the ratio between low-frequency and high-frequency power increased, whereas in episodes with hypotension, this ratio decreased [23].

Recently, in addition to the single-variable and hypothesis-based methods mentioned above, logistic regression using multivariate, negative binomial models, and deep learning (DL) methods have been studied for IDH prediction. In 2013, dialytic age, SBP, hemoglobin level, and weight gain factors during dialysis were found to be related to IDH in a multivariate regression model [24]. In 2018, a time-dependent logistic regression model calculated the BP drop probability, and the parameters used included SBP at the onset of IHD, current SBP, current dialysis setting, baseline demographic variables, and time lapse between the current time and last record [25]. In 2019, a multivariate negative binomial model using patient information and HRV for IDH prediction was proposed [13]. In addition, in 2020, a fully adjusted multivariate regression analysis and deep neural model using UF amount, UFR, UF coefficient, and comorbidity of hypertension were proposed based on a multi-factor interaction analysis [10]. Ref. [26] proposed a recurrent neural network using a timestamp-bearing dataset to predict IDH. To improve the performance of IDH prediction, a new feature selection method using Long Short-Term Memory was proposed in [27]. Ref. [28] also utilized time-related differences in machine learning and DL methods for IDH early warning. Table 1 presents a comparison of the IDH prediction models studied to date.

Currently, an important issue in IDH is medical response based on real-time detection. In the past, IDH studies used a large amount of static data, such as basic patient information, underlying disease, and the composition of compounds in the body during dialysis. Despite the recent active study of IDH prediction using DL and HRV information, there are still questions regarding the real-time application of these methods. For example, in [10], frequency HRV with a beat-to-beat (RR) interval for input data with a length of 240 min was used as an input for a DL model for IDH prediction; however, the real-time application of the method may not be easy because of the long input data lengths. In addition, as discussed in Table 1, in [13], HRV information as well as patient characteristic values were used as inputs to the DL model. However, the HRV values used in the method were representative values for time, and time-series values were not used as inputs. Therefore, this study aims at the real-time prediction and application of IDH.

In addition, our approach in this paper avoids the criticisms of existing methods using instantaneous data given in clinical practice on the doubt that some instantaneous data may not macroscopically reflect the occurrence of IDH. Specifically, it is proposed that changes in macroscopic pulse gradient as well as instantaneous pulse changes can be effectively utilized to predict abnormal BF such as IDH. In addition, a sudden change in BF, such as IDH, was used in the proposed model by discovering characteristics related to the body’s homeostasis, that is, the body’s responses aimed at reducing BF changes.

## 2. Pathogenesis and Medical Treatment for IDH

Figure 1 shows the normal and inadequate compensatory mechanisms for maintaining BP during dialysis via UF, as well as the actions of doctors for IDH. In the figure, the blue background represents venous circulation, whereas the red background represents arterial circulation. In the United States, the UF volumes for patients undergoing hemodialysis three times a week are typically in the range of 2.7–3.0 L [29].

As shown in Figure 1a [14], normal compensatory responses include activation of the sympathetic nervous system (SNS), increased release of the renin-angiotensin-aldosterone system and vasopressin (vasoconstrictor), and appropriate plasma replenishment (minimizing the decrease in blood volume in blood vessels) from the intracellular compartments. These responses promote the maintenance of BP by increasing cardiac preload through increased venous capacitive reflux, increased CO, and arteriolar vasoconstriction. BP is calculated as the product of CO and total peripheral resistance. CO is determined by stroke volume and HR, whereas stroke volume depends on the preload, afterload, and contractility. In response to hypotension, SNS stimulates increased HR and contractility to increase CO as well as BP. The various vasoactive hormones, including arginine vasopressin, SNS, and the renin-angiotensin-aldosterone system, increase total peripheral resistance to maintain the appropriate BP. Damage to any aspect of the normal compensation response, as shown in Figure 1b, may result in damage to the maintenance of the appropriate perfusion pressure, resulting in an IDH. It can also be assumed that the main factor of IDH is associated with the occurrence of intravascular blood loss.

### 2.1. Maintenance of Cardiac Output

#### 2.1.1. HR

A typical physiological response to hypovolemia is an increase in HR. The prevalence of tachyarrhythmia in patients undergoing hemodialysis is higher during dialysis. Drugs that cause negative inotropy can more easily induce IDH. On the other hand, slowing the HR and improving ventricular compliance can increase diastolic filling and minimize IDH.

#### 2.1.2. Contractility

Heart failure is an important risk factor for IDH, occurring in approximately one-third of patients undergoing hemodialysis [30]. In addition, >70% of dialysis patients show left ventricular hypertrophy during hemodialysis. Diastolic dysfunction can result in significant decreases in CO and BP because of small decreases in cardiac preload and left ventricular volume. Systolic failure has been reported in 15% of dialysis patients, and increasing cardiac contractility may also improve BP. High dialysate calcium levels may be used to increase myocardial contractility.

### 2.2. Cardiac Preload

The representative body response to hypovolemia is increased CO through increased cardiac preload. In hypovolemic conditions, an increase in vasoactive hormones and SNS results in arteriolar vasoconstriction and a reduction of blood flow to the venous beds. This lowers the pressure in the venous capacitance system and increases the venous return by causing subsequent passive elastic contraction of the vessel walls. This phenomenon is known as the DeJager-Kroger phenomenon [31]. Patients prone to hypotension during dialysis have a reduced DeJager-Kroger reflex response.

### 2.3. Arteriolar Vasoconstriction

#### 2.3.1. ANS

The development of peripheral arteriolar vasoconstriction in hypovolemia is regulated by the activity of the ANS and vasoactive hormones. Hypovolemia activates cardiopulmonary and baroreceptors, resulting in the inhibition of sympathetic outflow into the peripheral vasculature. This initially causes cutaneous arteriolar constriction followed by increased contractility and HR. Patients prone to hypotension during dialysis experience impaired sympathetic activation.

#### 2.3.2. Vasopressor Hormones

Inappropriate (blunted) elevation of vasoconstrictor hormones for hypovolemia is also associated with IDH. In particular, arginine vasopressin has a strong vasoconstrictive effect. Under normal conditions, arginine vasopressin is stimulated by severe hypovolemia and increased plasma osmolality. However, patients prone to hypotension during dialysis exhibit a blunted vasopressin response [32]. For this reason, vasopressin injection has been reported to be advantageous for the prevention of IDH.

The pathophysiology of IDH is shown in Figure 2 [33]. The body has various defenses against hypovolemia. These defenses include increased vascular resistance, which increases the HR and contractility. Reduced venous return is the most important factor that impairs the body’s ability to maintain CO. Diabetes, aging, and uremia can cause autonomic and baroreceptor dysfunction, leading to abnormal vasodilation.

Collectively, the common causes of IDH can be summarized as an excessive UFR, decreased CO, increased arteriolar tone, and autonomic dysfunction. In Figure 2, the green letters indicate the actions to be taken in response to the risk of IDH. Dialysate calcium, cold dialysate, and L-carnitine can be used to improve the heart function. Furthermore, to lower BP, midodrine may be administered or food intake may be prohibited. Additionally, to attenuate the variability that may lead to SBP reduction or IDH during dialysis, it may be helpful to use calcium channel blockers, discontinue α-blockers (antihypertensives), or maintain adequate serum phosphorus levels [34].

## 3. Materials and Methods

### 3.1. Participants

Data used to construct the neural networks were collected clinically from 80 hemodialysis patients in the artificial kidney room of Yeungnam University Medical Center (YUMC) in Daegu, South Korea (IRB File No.: YUMC 2019-04-035-001, YUMC 2020-08-005). The age of the patients has a distribution of 1 in their 30s, 3 in their 40s, 16 in their 50s, 34 in their 60s, 25 in their 70s, and 10 in their 80s. A total of 12 hemodialysis sessions (1–2 h per session) were performed during the data collection period (9 September 2020, to 17 October 2020). A study was conducted to evaluate the correlation between heart rate fluctuations and BP through bio-signals collected using a personal electrocardiogram (Vital Patch, VP-100) in hemodialysis patients [32]. Table 2 shows the statistical characteristics of the patients on renal dialysis who participated in the clinical trial. In addition, the R-peak detection methods of [35,36], and [37] were applied to electrocardiogram signals of hemodialysis patients for HR calculation.

### 3.2. Proposed Multilayer Perceptron Model for IDH Prediction

Figure 3 shows the structure of the multilayer perceptron (MP)-IDH net, which is a MP model proposed in this study for IDH prediction. The proposed model has a non-deep feedforward neural network structure, and its inputs are largely divided into static data and dynamic (time series).

The first static input data for the model was age, as used in previous studies such as [10,13] and [24,25]; elderly patients have a high risk of developing IDH. In addition, patients with comorbidities, such as diabetes ([12,13]) and hypertension ([10]), are more frequently observed to be prone to IDH. The last static data point is the most widely used UF amount.

The dynamic input data are the slope of the HR and the difference of slope (DoS) for 60 min, 30 min, and 15 min before the onset of IDH, as described in Section 3.2. The decrease in HR slope before IDH occurrence reflects the non-functioning of the compensatory response of sympathetic increase in heart rate to IDH-induced hypovolemia. The decrease in the HR slope before the occurrence of IDH reflects the non-functioning of the compensation response of increased HR by sympathetic nerves to hypovolemia caused by IDH. These HR slopes represent the macroscopic phenomenon of IDH occurrence. The last dynamic input data is the time from the measurement time to point E, representing the macroscopic phenomenon of IDH generation. This reflects an increase in HR due to sympathetic nerve activation, which often occurs before hypotension during dialysis. The proposed MP structure has 10 inputs, one or two hidden layers, and one output layer. The input and hidden layers have the same number of 10 nodes.

## 4. Results

### 4.1. Changes in HR Slope before IDH

Figure 4 shows the changes in the slope of the HR per minute for 15 min (blue line), 30 min (green line), and 1 h (red line) before the onset of IDH. In addition, the black star indicates the point with the largest difference in HR for 1 h before the onset of IDH, which is referred to as the emergency point (point E) in this study. As shown in the figure, most of the ECG signals (Figure 4a–h) showed a tendency to decrease the average HR per minute slope. In addition, for some ECG signals (Figure 4i,j), a slight increase in the heart rate per minute gradient was observed. As introduced in Section 2, the expression of point E in the proximate time period before the occurrence of IDH can be considered to be due to an increase in HR for an increase in BP by the SNS for hypovolemia caused by hypotension during dialysis.

It can be seen that this instantaneous increase in HR per minute is more pronounced in Figure 4c–e,g,h. Point E is an important parameter in IDH prediction. The E-point, which represents a momentary increase in HR, provides significant evidence for the occurrence and prediction of IDH, but the average HR slope obtained for a specific time period before the onset of IDH does not have this E-point information. Therefore, E-point information is independently used as an input to the proposed MP model. In a significant portion of our clinical data, a decrease in HR slope was observed closer to the occurrence of IDH. This lowering of the HR slope suggests that the compensatory response of increased HR to IDH-induced hypovolemia does not work well in patients with IDH. Therefore, changes in the HR slope can be an important factor in IDH prediction. Collectively, it can be observed that point E represents the instantaneous phenomenon of IDH generation, and the change in the HR slope represents the macroscopic phenomenon of IDH generation.

### 4.2. Analysis of E-Point

HR is largely regulated by the ANS, including the sympathetic and parasympathetic nervous systems [38]. SNS increases HR while the parasympathetic nervous system (PNS) suppresses it. Figure 5 shows the homeostasis (constancy) of BP based on the relationship between BP and baroreceptor reflexes [39]. In hypotensive conditions, such as IDH, baroreceptor firing is reduced, resulting in a decrease in cardiac inhibitors and activation of cardiac accelerator and vasomotor centers. Thereafter, BP increases, owing to an increase in CO (HR) and a decrease in vasoconstriction [39,40]. This BP homeostasis results in the expression of point E, as discussed in Section 3.2.

As introduced in Section 3.2, the expression of point E can be attributed to the increase in HR for increased BP due to homeostasis of the SNS for hypovolemia caused by hypotension during dialysis. In this study, it is assumed that the precursor phenomenon before the occurrence of IDH, named point E, occurs in the HR difference. Figure 6 shows the distribution of HR differences in patients with IDH and non-IDH (normal) 1 h before the occurrence of IDH. Under the tendency of hypotensive variability in patients with IDH (cf. the HR slope before IDH in Figure 4), each E point is associated with the body’s homeostasis by the activation of sympathetic nerves against tachycardia that occurs in individual cases. That is, the reaction indicates a temporary increase in HR by the sympathetic nerve to raise the lowered HR. As shown in Figure 6, in patients with IDH, 60% of all E-points appeared within 30 min of IDH onset, whereas in normal (non-IDH) patients, the E-points were evenly distributed for 1 h before IDH onset.

### 4.3. Analysis of HR Slope

Figure 7 and Table 3 show the HR slopes of the 1 h, 45 min, 30 min, and 15 min data before IDH onset for IDH and normal (Non-IDH) patients. Each HR slope was obtained using the line-fitting function in MATLAB R2021b for a given time period before the occurrence of IDH. In each HR slope plot for each time period, the bold color line indicates the average HR slope for the corresponding time period. It can be seen that the mean HR slope decreases as we approach the IDH generation.

This section shows the correlations between diabetes, hypertension, age, hyperfiltration and IDH incidence through HR slope analysis. Figure 8 shows the changes in HR slopes for IDH and normal (non-IDH) patients according to 30-min data for each patient’s baseline information before IDH onset (see Appendix A for graphs of the 60 min and 45 min data prior to the onset of IDH). The HR slope values for Figure 8 are listed in Table 4. Data at 30 min before IDH onset were more dramatic than those at 60 and 45 min before the onset of IDH (see Appendix A for slope values of the 60 min and 45 min data prior to the onset of IDH).

Regardless of the presence or absence of diabetes, it can be seen that the HR slope decreases as IDH development approaches. In the case of non-diabetic patients, when IDH occurs, it can be seen that the HR slope was significantly reduced compared to the normal (non-IDH) case. In contrast, in the case of diabetic patients, the HR gradient gradually decreased as IDH development approached. In the absence of IDH, there was little change in the HR slope, with or without diabetes (nearly zero HR slope). This finding implies that changes in the HR slope in diabetic patients affect IDH occurrence.

In the absence of IDH, there was little change in the HR slope, with or without hypertension. That is, the HR slope of hypertensive patients gradually decreased, whereas the HR slope of non-hypertensive patients slightly increased. Similar to the occurrence of IDH in diabetic patients, in the case of hypertension, the HR slope decreased as IDH development approached. Similar to IDH development in diabetic patients, it was confirmed that changes in the HR slope in hypertensive patients affect IDH occurrence.

As the onset of IDH approached, the HR slope of elderly patients aged 65 years and older became much smaller than that of the people under 65 years of age. This implies that old age influences the development of IDH. In addition, when IDH did not occur, the HR slope of the elderly patients decreased slightly, whereas that of the non-elderly people remained almost unchanged.

In the case of a UF amount of 2000 or more, the HR slope decreases as IDH generation approaches, whereas in the case of a UF amount less than 2000, the HR slope hardly changes, even as IDH generation approaches. In addition, when IDH did not occur, a UF amount over 2000 showed a constant change in HR slope, whereas a UF amount less than 2000 induced an increase in HR.

### 4.4. Model Results

The MP structure proposed in this study for predicting hypotension during dialysis was implemented using the DL toolbox of MATLAB R2021b. The MP architecture used had 10 input neurons, two neurons per hidden layer, and 10 output layer neurons. In addition, the MP structure used a scaled conjugate gradient method and a cross-entropy cost function, and typically 10–30 epochs were used.

The performance of the proposed model was evaluated in terms of accuracy (ACC), sensitivity (e.g., true positive rate (TPR)), precision (e.g., positive predictive value (PPV)), and Matthews correlation coefficient (MCC), which were calculated from true positive (TP), true negative (TN), false positive (FP), and false negative (FN) values. ACC, TPR, PPV, and MCC were calculated as ACC = (TP + TN)/(TP + FP + TN + FN), TPR = TP/(TP + FN), PPV = TP/(TP + FP), and MCC = {(TP × TN) – (FN × FP)}/sqrt {(TP + FN) (TN + FP) (TP + FP) (TN + FN)}, respectively [10].

Figure 9 shows the confusion matrix of the training and test data for Deep-IDH models using different hidden layers and data lengths. The first two diagonal cells in the figure show the number and percentage of correct classifications using the trained network. The rows correspond to the predicted classes (output classes), and the columns correspond to the actual classes (target classes). Cells outside the main diagonal correspond to misclassified observations [41]. The right column of the plot shows the precision (PPV) and FN rate, the lower column shows the recall (TPR) and FN rate, and the cell in the lower right of the plot shows the overall accuracy [42]. In training using one layer, the model accuracy using 60-min data before IDH occurrence was 69.2%, whereas the model accuracy using 45-min data before IDH occurrence was 73.2%. In training using two layers, the model accuracy using 60-min data before IDH was 81.5%, whereas the model accuracy using 45-min data before IDH was 70.2%. Regardless of the number of layers, the model using 30-min data showed lower accuracy than the models using longer length data. This means that the minimum time to extract the HR slope information for IDH prediction is 45 min or more.

Table 5 lists the performance of Deep-IDH models using different hidden layers and data lengths. The model using the 2-layer hidden layer and 60 min data before the occurrence of IDH showed the maximum value after the performance metrics evaluation [ACC = 81.5 (%); TPR = 73.8 (%); PPV = 87.3 (%); and MCC = 0.638]. The MP-IDH model using one layer and 60 min data before IDH occurrence showed the maximum TPR value, and the longer the data length for the same number of hidden layers, the better the performance. In addition, the model using a data length of less than 60 min (30-min and 45-min data) before the occurrence of IDH showed higher ACC, PPV, and MCC values, as the number of hidden layers decreased. This suggests that reducing the number of hidden layers results in a better performance when the data length is less than 1 h.

The efficiency of the proposed Deep-IDH model was analyzed using a receiver operating characteristic (ROC) curve. Figure 10 shows the ROC curves of the Deep-IDH models using different hidden layers and data lengths. ROC curves show TPR (sensitivity) versus FPR (1-specificity) for various thresholds of the classification scores. As shown in the figure, it can be seen that the ROC curve of the Deep-IDH model using two layers and 60-min data before IDH occurrence is on top of the ROC curves of other models, and can work as a better binary classifier for IDH prediction. As shown in Figure 10c,f, it can be seen that the shorter the length of available data before IDH occurrence, the closer the ROC curve to the mean/basic model. In addition, as described in Table 5, when data less than 60 min before the occurrence of IDH have to be used, the smaller the number of hidden layers, the better the binary classifier (note the change from Figure 10e,f to Figure 10b,c).

This study verified a MP system for real-time prediction of IDH occurrence. Predicting the BP profile during a 1–2 h hemodialysis session is a difficult task. The Deep-IDH model we built performed well for various definitions of SBP drop, especially for <90 and 100 mmHg, which were most associated with mortality among the definitions of IDH in previous studies [25,29]. This indicates that our predictive model is stable and reliable for predicting IDH in hemodialysis patients.

## 5. Discussion

### 5.1. ECG Analysis of Patients with IDH

Figure 11 shows the HRV results at the time of hypotension in five hemodialysis patients using the HRV analysis method introduced in [43,44]. Among the HRV parameters, HR per minute (beats per minute (BPM)), RR interval trend, RR histogram, power spectrum, and Poincaré plot were used. The RR interval refers to the elapsed time (reciprocal of HR) between two consecutive R peaks in the ECG. The power spectral density (PSD) is estimated from the RR series using the fast Fourier transform, and all frequency domain parameters of the PSD are calculated according to the specified frequency band. The Poincaré plot also shows a scatterplot of the current RR interval plotted against the previous RR interval; it can be seen that the characteristics are prominent at the time of hypotension in hemodialysis patients. It can also be seen that the higher the instantaneous change rate of HR per minute and RR interval, or the higher the non-clusterity in the Poincaré plot, the higher the probability of hypotension. However, it is difficult to use these characteristic values in real-time because of the large instantaneous change rate.

Figure 12 shows the change in HR per minute at the onset of IDH in patients undergoing hemodialysis. In the figure, the orange square box indicates the IDH or hypotension period and includes the systolic and diastolic BP values. In addition, the blue line represents the original HR per minute value and the red line represents the average HR per minute value. For patients #1 and #4 with normal HR per minute, the change in HR per minute was very severe (20 or higher HR per minute) from 30 min to 1 h before the onset of hypotension. In addition, there were many sections where the average HR per minute changed significantly. In patient #3 with bradycardia (HR less than 70 in clinical practice), hypotension occurred after the HR per minute gradually decreased for > 30 min. That is, when the HR per minute decreased gradually in patients with bradycardia or when the HR per minute fluctuated in patients with normal HR per minute, the probability of occurrence of hypotension was high. To generalize such instantaneous changes in HR per minute, we analyzed the slope of HR per minute for a specific time period before IDH, as shown in Figure 4.

### 5.2. Real-Time Prediction of IDH

A major issue in current IDH research is real-time prediction. Hospital officials have demanded over 65% real-time accuracy for IDH prediction. The IDH prediction methods studied thus far have an accuracy of 65% or more. However, the major requirement is that it should be capable of real-time processing. Until now, many IDH prediction methods using static data faced difficulties when applied in real-time. In addition, even if it is a method that uses time-series data, the methods used very long data (approximately 4 h), and representative values of the used data length were used as input to the MP model. As such, the data length for real-time prediction of IDH is judged to be appropriately within 30 min to 1 h or 2 h before IDH occurs. In addition, although the current study used data up to the onset of IDH, considering the amount of change in the slope of HR, it is judged that data up to 5 min before the onset of IDH can be used for practical use.

In this study, a real-time IDH prediction model with an ACC of 81.5% and a PPV of 87.3% was created using patient characteristics such as age, diabetes, and hypertension, and a relatively limited number of parameters, such as UF volume and pulse information. The BPM slope and symptom points can be easily calculated in real time. Because we used a relatively limited number of predictors in this study, future models with a higher level of segmentation may perform better.

### 5.3. Data and Clinical Issues of IDH

Another issue is the lack of open databases for IDH predictions. Because the data obtained through clinical trials are used for each study, the current IDH prediction study encounters the large problem of data individuality. In addition, small IDH trials may have data bias issues. Owing to these practical problems, it is difficult to objectively compare the IDH prediction methods studied to date. Therefore, the selection of standardized parameters necessary for IDH prediction is essential for future research, and an objective performance comparison of IDH prediction techniques developed so far can be performed afterwards.

Table 6 shows the parameters for patient data and session data that can be considered in addition to the factors used for IDH prediction introduced in Table 1 [45,46]. There may also be other parameters that could be potential factors for real-time IDH prediction. The use of different parameters by different investigators may be because of the clinical environment. Therefore, it can be a good choice to set up the clinical environment after selecting the necessary parameters through sufficient review of existing studies.

## 6. Conclusions

In this study, a method for predicting IDH in real-time was introduced, and the pulse information extracted from the ECG signal was used as the time-series data input for the MP model. Considering the pulse information used in the MP model, it was inferred that the change in the pulse slope before IDH occurrence and the appearance of symptom points were highly correlated with the occurrence of IDH. In addition, not only time-series data, but age, diabetes, hypertension, and UF, which are static data used in many IDH prediction methods studied so far, were also used as inputs for the MP model proposed in this study. This study aimed to predict IDH, which may occur in a short time, in real time by continuously inputting real-time pulse information into a MP model. Our approach avoids the batch use of instantaneous data, which may have a low correlation with a given IDH in a clinical setting, as found in other studies. Specifically, it is based on the inference that changes in BF lead to changes in the macroscopic slope of the pulse; we found that sudden abnormal BF reactions such as IDH lead to various characteristic responses due to homeostasis of the body, and visible as symptom points. In this study, we implemented a MP prediction model using relatively limited parameters, and we believe that the model’s performance can be further improved through the future study of additional parameters.

## 7. Patents

A patent from the results of this study is pending approval. (Title: *Method for predicting hypotension during dialysis using patient baseline information and heart-rate variation.*).

## Figures and Tables

**Figure 1 ijerph-19-10373-f001:**
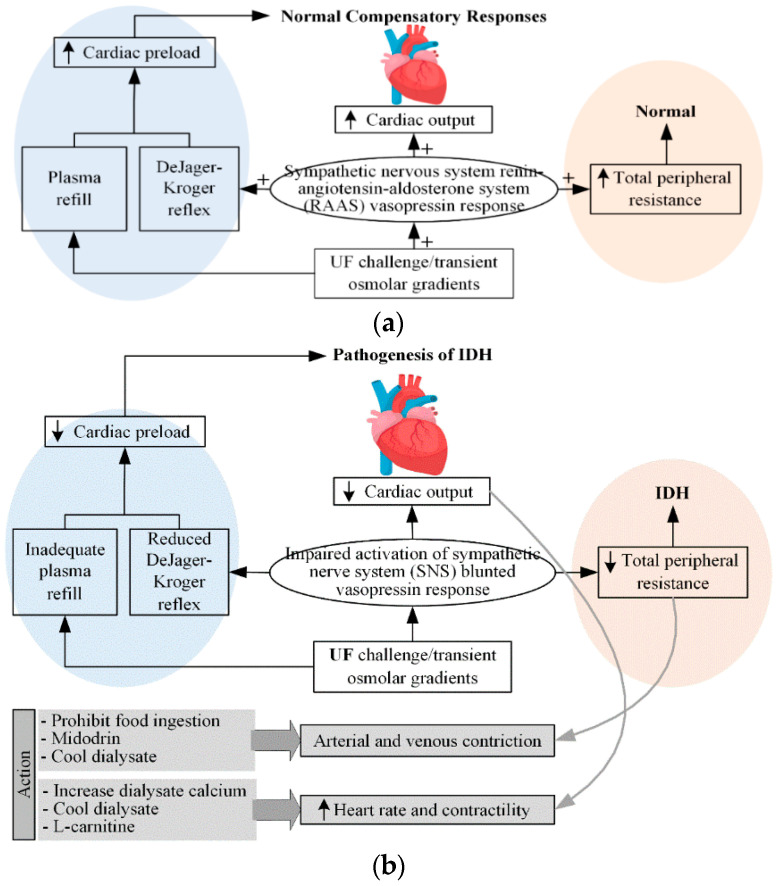
(**a**) normal and (**b**) inadequate compensatory responses to maintain BP during dialysis ([14]). (In Figure 1a, + denotes the response sensitivity.)

**Figure 2 ijerph-19-10373-f002:**
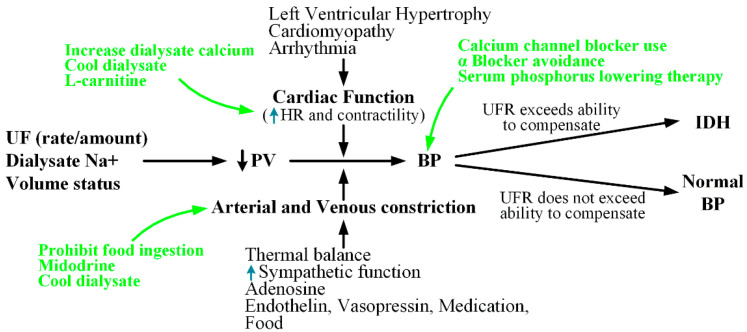
Pathophysiology of IDH ([33]).

**Figure 3 ijerph-19-10373-f003:**
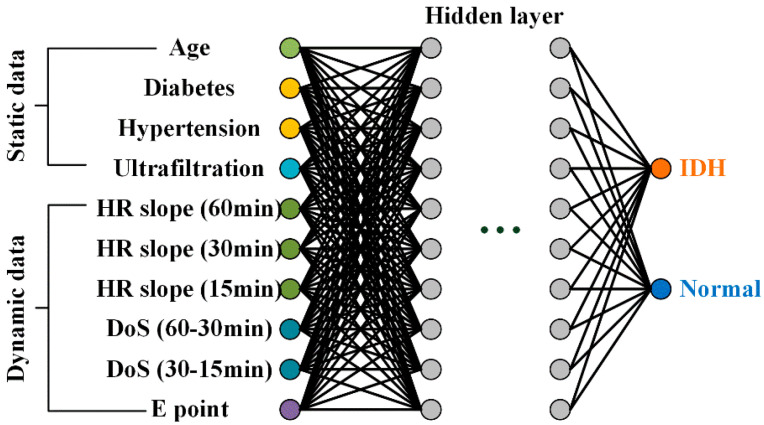
Proposed MP-IDH net with static and dynamic data inputs.

**Figure 4 ijerph-19-10373-f004:**
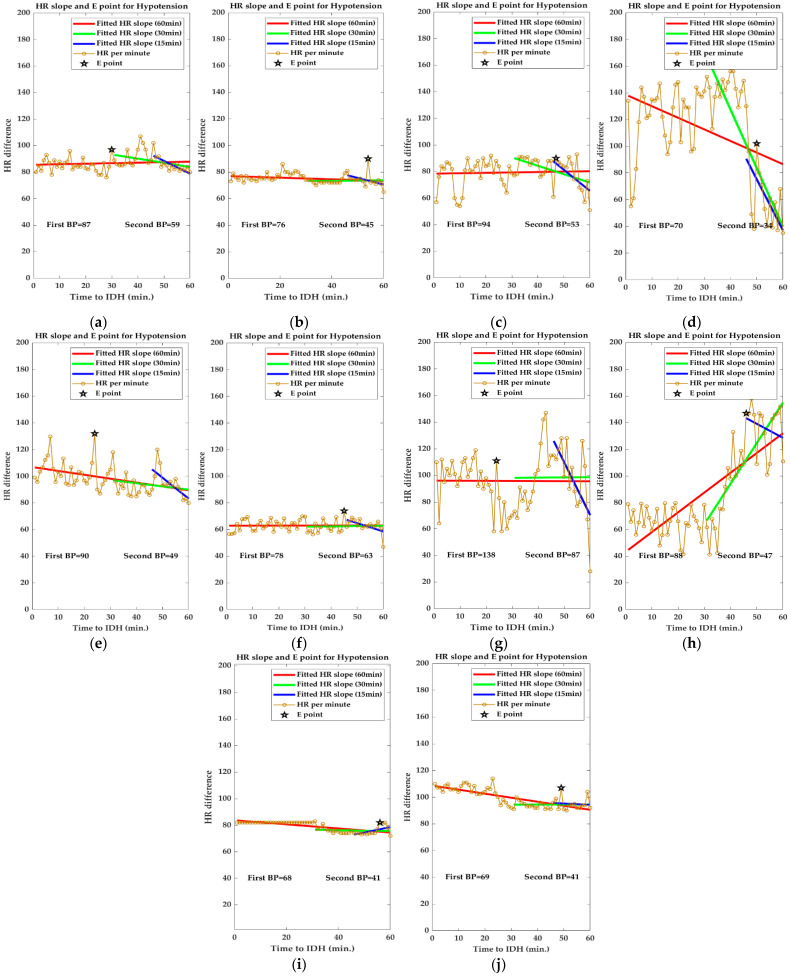
Changes in slope of HR per minute before IDH. ((**a**–**h**,**I**,**j**) show the decreasing and increasing trend of the mean HR slope respectively).

**Figure 5 ijerph-19-10373-f005:**
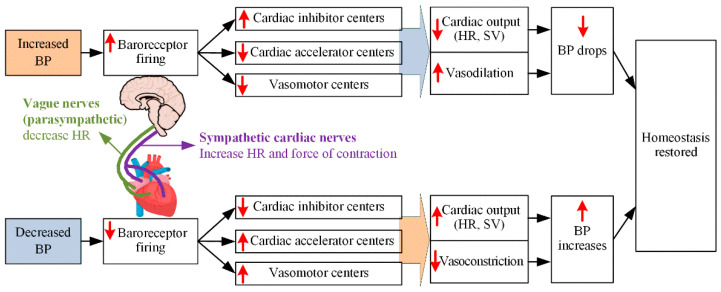
Relationship between BP and baroreceptor reflex ([39]).

**Figure 6 ijerph-19-10373-f006:**
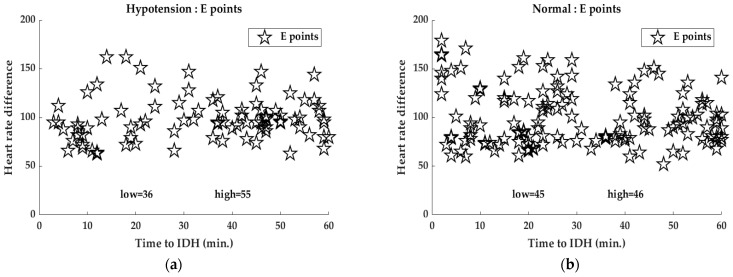
Distribution of HR differences for (**a**) IDH and (**b**) normal (non-IDH) patients 1 h before IDH onset.

**Figure 7 ijerph-19-10373-f007:**
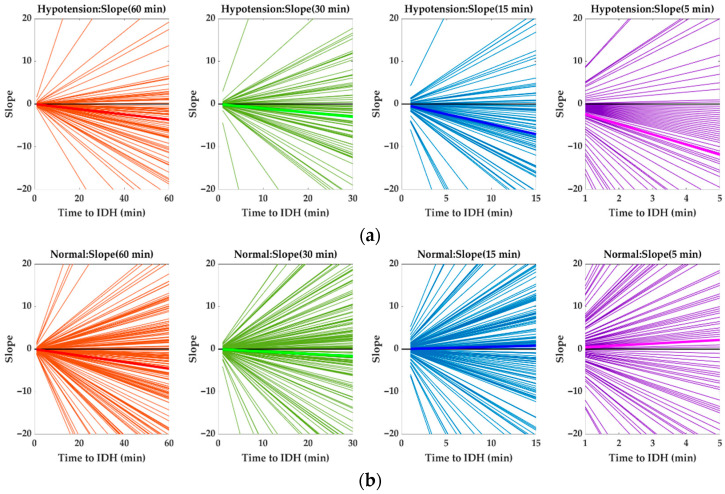
Changes in HR slopes of 1 h, 45 min, 30 min, and 15 min data for IDH and normal (Non-IDH) patients before IDH onset. HR slopes before IDH onset for (**a**) IDH and (**b**) normal (Non-IDH) patients. (bold color line: average HR slope, black line: *x*-axis).

**Figure 8 ijerph-19-10373-f008:**
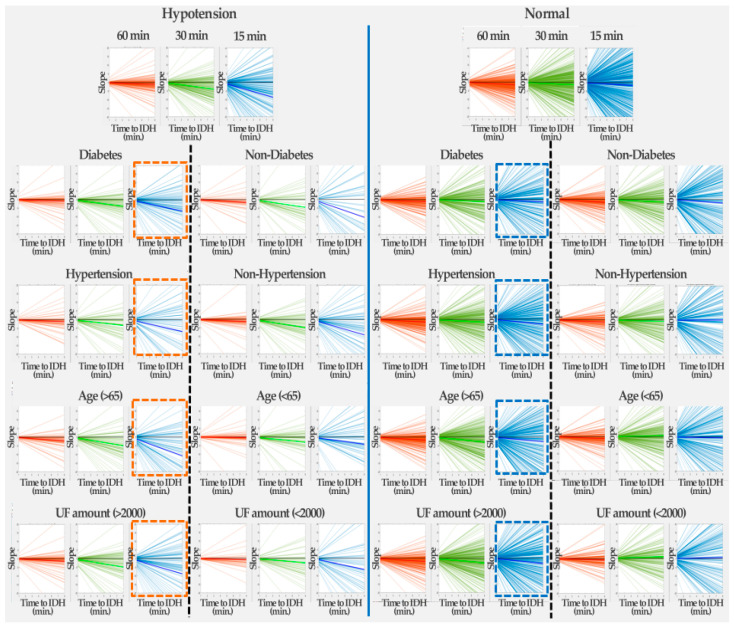
Changes in HR slope for IDH and normal (non-IDH) patients according to 30-min data by patient baseline information before the onset of IDH. (The dashed lines represent the lines separating the patient baseline information).

**Figure 9 ijerph-19-10373-f009:**
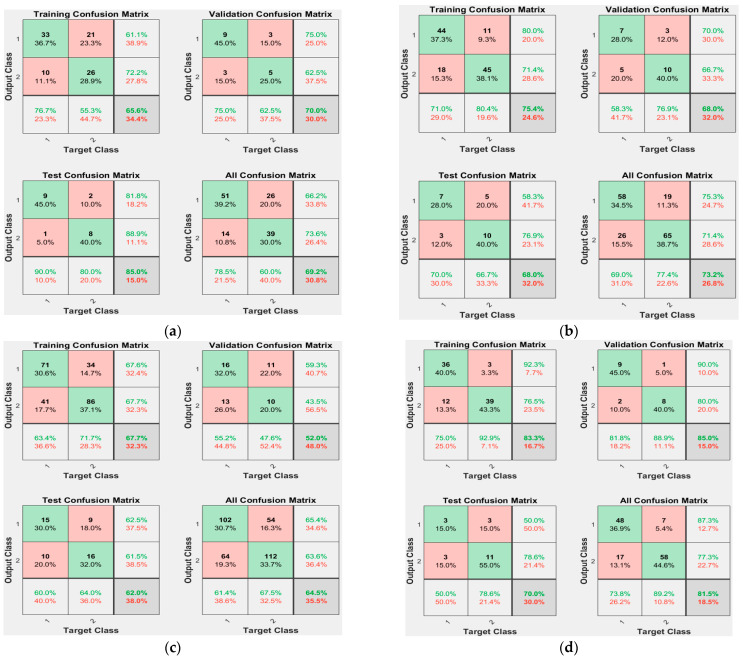
Confusion matrices of Deep-IDH models using different hidden layers and data lengths: (**a**) 1-layer and 60-min data before IDH occurrence (69.2%, 30.8%); (**b**) 1-layer and 45-min data before IDH onset (73.2%, 26.8%); (**c**) 1-layer and 30-min data before IDH onset (64.5%, 35.5%); (**d**) 2-layer and 60-min data before IDH onset (81.5%, 18.5%); (**e**) 2-layer and 45-min data before IDH onset (70.2%, 29.8%); and (**f**) 2-layer and 30-min data before IDH onset (59.6%, 40.4%).

**Figure 10 ijerph-19-10373-f010:**
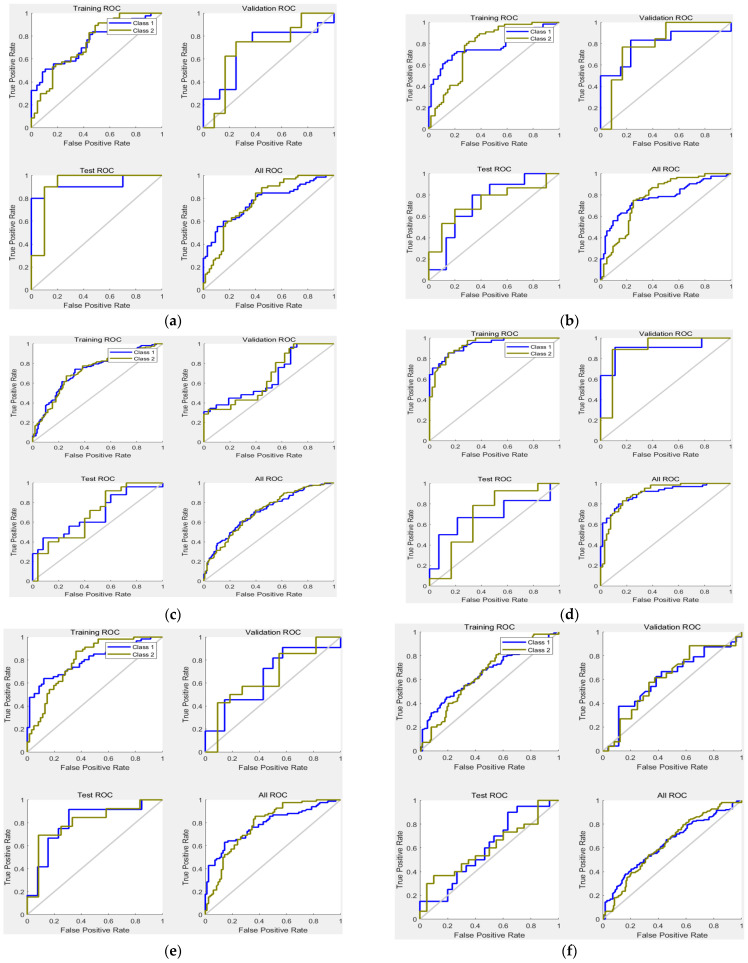
ROC of Deep-IDH models using different hidden layers and data lengths: (**a**) 1-layer and 60-min data before IDH onset; (**b**) 1-layer and 45-min data before IDH onset; (**c**) 1-layer and 30-min data before IDH onset; (**d**) 2-layer and 60-min data before IDH onset; (**e**) 2-layer and 45-min data before IDH onset; and (**f**) 2-layer and 30-min data before IDH onset.

**Figure 11 ijerph-19-10373-f011:**
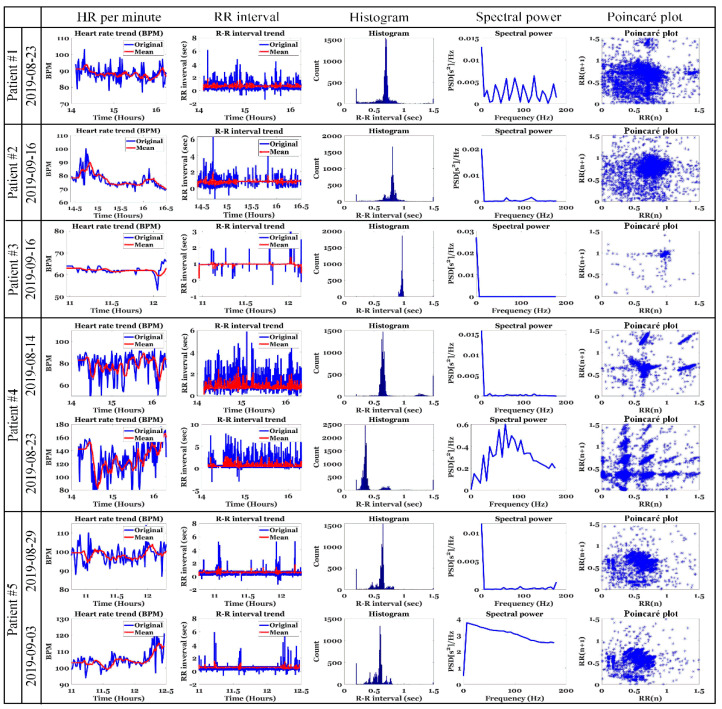
HRV results at the time of hypotension in hemodialysis patients.

**Figure 12 ijerph-19-10373-f012:**
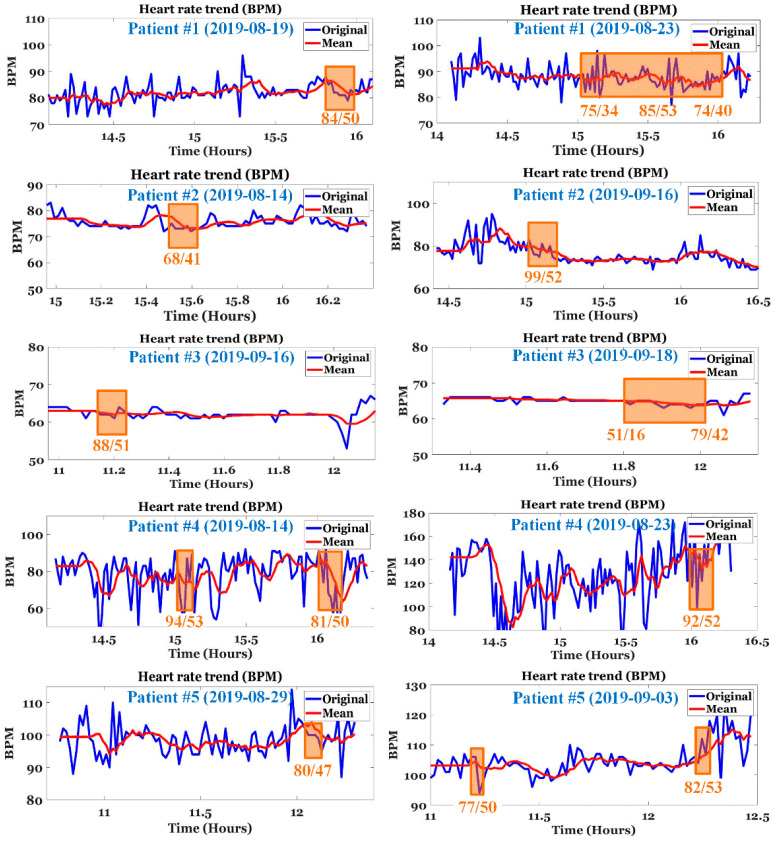
Changes in HR per minute at the onset of IDH in hemodialysis patients. (The square box area indicates the IDH or hypotension period).

**Table 1 ijerph-19-10373-t001:** Comparison of IDH prediction models developed to date.

References	Factors Used	Model Used	Data Source	Performance
Solem et al. [19], 2010	Amplitude of PPG	Hypothesis based statistical model	11 patients 25 treatments	57~65%
Bossola et al. [24], 2013	Age, Sex, CCIS, Hemoglobin, Serum creatinine, Serum albumin, DSC, Blood flow, IWG, ACE-Inhibitors or Sartans, Predialysis SBP, Dialytic age	Linear and logistic regression model	82 patients	-
Sandberg et al. [9], 2014	PPG envelope, LF/HF ratio of ECG	Bayes’ rule	28 sessions from 11 hypotension-prone patients, 20 sessions from 7 patients	9/14 (symptomatic IDH), 5/5 (acute symptomatic IDH)
Shahabi et al. [22], 2015	Time domain features and LF/HF ratio of PPG	Genetic algorithm and AdaBoost	10 patients 217 Normal, 22 Pre-IDH episode, 90 IDH episode	Accuracy of 90.68 %)
Lin et al. [25], 2018	- SBP at onset, current SBP, Time lapse to next SBP - Dialysis settings (machine temperature, Conductivity, UFR) - Baseline demographic variables (age, sex, diabetes mellitus, dry weight)	Time-dependent logistic regression model	653 HD outpatients, 55,516 HD treatment sessions	Sensitivity of 86% and specificity of 81%
Park et al. [13], 2019	Diabetes mellitus, CAD, CHF, Age, UFR, iPTH, ARB, CCB, β-blocker, RRI, HF, TP, AIC	Multivariate negative binomial model	28 patients, 85 cases (10% of a total 852 dialysis sessions)	-
Chen et al. [10], 2020	Age, BMI, Gender, Comorbidity of hypertension, UF coefficient, UF amount, UFR, Ca, Cardiothoracic ratio	Deep Neural Network	279 participants, 780 hemodialysis sessions	-
	Comorbidity of hypertension, UF coefficient, UF amount, UFR	Deep Neural Network		
Lee et al. [26], 2021	Age, Male, Hemodialysis type, Vascular access, Anticoagulant, Blood findings, Dialysate finding	Recurrent Neural Network	9292 patients, 261,647 sessions	AUC of 0.94
Hu et al. [27], 2021	Blood draw data, Physiological measurement data, Time series	Long Short-Term Memory	593 dialysis sessions	AUC of 0.97
Yang et al. [28], 2021	Time-relevant difference	Light Gradient Boosting Machine	593 hemodialysis sessions	Sensitivity of 88.9%

CCIS, Charlson comorbidity index score; DSC, dialysate sodium concentration; IWG, interdialytic weight gain; CAD, coronary artery disease; CHF, congestive heart failure; UFR, ultrafiltration rate; iPTH, intact parathyroid hormone; ARB, angiotensin II receptor blocker; CCB, calcium channel blocker; RRI, R-R interval; HF, high frequency; TP, total power; AIC, Akaike information criterion; AUC, area under the receiver operating characteristic curve.

**Table 2 ijerph-19-10373-t002:** Baseline characteristics of the patients in IDH and non-IDH groups.

	Total	Male, n (%)	Female, n (%)	Age > 65 y, n (%)	Age < 65 y, n (%)	Diabetes, n (%)	Non-Diabetes, n (%)	Hypertension, n (%)	Non-Hypertension, n (%)	UF Amount
Total	89	48 (53.9)	41 (41.6)	52 (58.4)	37 (41.6)	47 (52.8)	33 (37.1)	51 (57.3)	38 (42.7)	2182.7
IDH	67	30 (44.8)	37 (55.2)	41 (61.2)	26 (38.8)	35 (52.2)	28 (41.8)	29 (43.3)	29 (43.3)	2127.4
Non-IDH	22	18 (81.8)	4 (18.2)	11 (50.0)	11 (50.0)	12 (54.5)	5 (22.7)	9 (40.9)	9 (40.9)	2350.9

The units y and n mean year and number respectively.

**Table 3 ijerph-19-10373-t003:** HR slope values of 1 h, 45 min, 30 min, and 5 min data for IDH and normal (Non-IDH) patients before onset of IDH.

	Hypotension				Normal			
	60 min	30 min	15 min	5 min	60 min	30 min	15 min	5 min
Mean slope	−0.0608	−0.0987	−0.4706	−2.3681	−0.0764	−0.0555	0.0526	0.4310
Num. of positive slopes	25	33	24	18	73	103	108	95
Num. of negative slopes	66	58	67	73	111	81	76	89
% of negative slopes	72.5	63.7	73.6	80.2	60.3	44.0	41.3	48.4

**Table 4 ijerph-19-10373-t004:** HR slope values for IDH and normal (non-IDH) patients according to patient baseline information 30 min before IDH onset.

	Hypotension						Normal					
	30 min	15 min	8 min	30 min	15 min	8 min	30 min	15 min	8 min	30 min	15 min	8 min
Underlying disease	Diabetes			Non-diabetes			Diabetes			Non-diabetes		
Mean slope	−0.136	−0.489	−0.808	−0.207	−0.558	−1.303	−0.159	−0.155	−0.179	−0.224	−0.185	−0.259
Num. of positive slopes	27	21	44	13	16	32	94	132	209	85	127	201
Num. of negative slopes	76	82	59	50	47	31	261	223	146	258	216	142
% of negative slopes	73.79	79.61	57.28	79.37	74.60	49.21	73.52	62.82	41.13	75.22	62.97	41.40
Underlying disease	Hypertension			Non-hypertension			Hypertension			Non-hypertension		
Mean slope	−0.154	−0.429	−0.862	−0.171	−0.587	−0.109	−0.198	−0.215	−0.374	−0.182	−0.104	0.007
Num. of positive slopes	17	18	37	23	19	39	93	145	233	86	114	177
Num. of negative slopes	59	58	39	67	71	51	320	268	180	199	171	108
% of negative slopes	77.63	76.32	51.32	74.44	78.89	56.67	77.48	64.89	43.58	69.82	60	37.89
Underlying disease	Age			Non-age			Age			Non-age		
Mean slope	−0.254	−0.6198	−1.353	−0.045	−0.379	−0.529	−0.221	−0.357	−0.332	−0.151	0.087	−0.064
Num. of positive slopes	14	16	42	26	21	34	89	116	242	90	143	168
Num. of negative slopes	80	78	52	46	51	38	314	287	161	205	152	127
% of negative slopes	85.11	82.98	55.32	63.89	70.83	52.78	77.92	71.22	39.95	69.49	51.53	43.05
Underlying disease	UF			Non-UF			UF			Non-UF		
Mean slope	−0.190	−0.607	−1.094	−0.107	−0.325	−0.792	−0.240	−0.335	−0.377	−0.103	0.134	0.072
Num. of positive slopes	27	23	54	13	14	22	101	149	253	78	110	157
Num. of negative slopes	85	89	58	41	40	32	351	303	199	168	136	89
% of negative slopes	75.89	79.46	51.79	75.93	74.07	59.26	77.65	67.04	44.03	68.29	55.28	36.18

**Table 5 ijerph-19-10373-t005:** Performance of Deep-IDH models using different hidden layers and data lengths.

Combinations	Model Ranking	ACC (%)	TPR (%)	PPV (%)	MCC
1 layer, 60 min	4	69.2 %	78.5 %	66.2 %	0.391
1 layer, 45 min	2	73.2 %	69.0 %	75.3 %	0.496
1 layer, 30 min	5	64.5 %	61.4 %	65.4 %	0.290
2 layers, 60 min	1	81.5 %	73.8 %	87.3 %	0.638
2 layers, 45 min	3	70.2 %	66.7 %	71.8 %	0.370
2 layers, 30 min	6	59.6 %	68.1 %	58.2 %	0.195

**Table 6 ijerph-19-10373-t006:** Additional parameters that may be considered.

Patient Data	Session Data
Male sex, Dialysis vintage, Race (White, Black), Peripheral artery disease, Peripheral vascular disease, Antihypertensive use, Body temperature	Interdialytic weight gain, Blood flow, Dialysate temperature, Dialysate conductivity, Dialysate sodium, Dialysate calcium, Body weight before and after HD

## Data Availability

Not applicable.

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
