# Peer review of "Multilayer Perceptron-Based Real-Time Intradialytic Hypotension Prediction Using Patient Baseline Information and Heart-Rate Variation"

_ijerph, 2022, doi:10.3390/ijerph191610373_

Round 1
Reviewer 1 Report
I think this is an interesting work for the early prediction of medical emergency leveraging machine learning method, and may raise interests from practitioners of hemodialysis. But I'm concerned with the practicality of the proposed method since the input can only be obtained until IDH happened when it's too late. Besides, I have the following questions.
1. Basic information of patient samples need to be introduced in M&M, section 3.1 can be rearranged.
2. Jargons of RR interval trend, RR histogram, power spectrum, and poincaré plot should be explained in figure legends. Besides, legends are missing in Fig.1-10.
3. As the author said in Line 246, the model develop is "non-deep" considering the size of the neuron network, so "Deep learning" is not appropriate to appear in the title. Terms like "multilayer perceptron" can be used instead.
4. Considering the number of features and samples in the training dataset, did the author tried other classifier like SVM, LR or random forest?
5. Incorrect numbering of Results.
6. "Figure 7" in Line 300 and 306 should be Figure 8.
7. What's the unit of x/y axis in Fig.9 and 10?
Author Response
Reviewer #1
I think this is an interesting work for the early prediction of medical emergency leveraging machine learning method, and may raise interests from practitioners of hemodialysis. But I'm concerned with the practicality of the proposed method since the input can only be obtained until IDH happened when it's too late. Besides, I have the following questions.
Ans) I had the same thoughts as you. Considering the amount of change in the slope of HR (heartrate), it is judged that data up to 5 minutes before the onset of IDH can be used. We have added this explanation in section 5.2, line 463 of the discussion.
----------------------------------------------------------------------------------------
In addition, although the current study used data up to the onset of IDH, considering the amount of change in the slope of HR, it is judged that data up to 5 minutes before the onset of IDH can be used for practical use.
- Basic information of patient samples need to be introduced in M&M, section 3.1 can be rearranged.
Ans) As per your advice, patient information has been moved to Section 3.1.
- Jargons of RR interval trend, RR histogram, power spectrum, and poincaré plot should be explained in figure legends. Besides, legends are missing in Fig.1-10.
Ans) As per your advice, on page 16, line 424, descriptions of the legends have been added as shown below. We've also added missing legends to each figure (especially Figures 7 and 8). We also improved the lower resolution of Figures 7 and 8.
----------------------------------------------------------------------------------------
The RR interval refers to the elapsed time (reciprocal of HR) between two consecutive R peaks in the ECG. The power spectral density (PSD) is estimated from the RR series using the Fast Fourier Transform, and all frequency domain parameters of the PSD are calculated according to the specified frequency band. The Poincaré plot also shows a scatterplot of the current RR interval plotted against the previous RR interval.
- As the author said in Line 246, the model develop is "non-deep" considering the size of the neuron network, so "Deep learning" is not appropriate to appear in the title. Terms like "multilayer perceptron" can be used instead.
Ans) As per your advice, the term Deep Learning has been modified to Multilayer Perceptron for both the title and the entire paper.
- Considering the number of features and samples in the training dataset, did the author tried other classifier like SVM, LR or random forest?
Ans) Yes, I was considering using the classifier you mentioned (SVM, LR or random forest). However, since two types of data (static patient information and dynamic HR sequential data) are used in my study, I decided that the multilayer perceptron (or shallow learning) method was slightly better than the existing classifier. We will consider your comments in future research. Thank you
- Incorrect numbering of Results.
Ans) Thank you, the numbering of Results has been corrected.
- "Figure 7" in Line 300 and 306 should be Figure 8.
Ans) The error has been corrected. Thank you.
- What's the unit of x/y axis in Fig.9 and 10?
Ans) The x-axis means the length of time before IDH occurs, and the Y-axis means the slope value. The legends have been added to Figure 7 and Figure 8, and the resolution of the figures has also been modified. Thank you.
Thank you for your valuable review !!

Reviewer 2 Report
SUMMARY
This work aims to predict intradialytic hypotension (IDH) using a deep learning approach fed mainly with risk-related parameters and continuous monitoring of heart rate. The authors highlight the importance of a fast and effective prediction of IDH, which affects a 15-20 % of patients undergoing dialysis and clearly represents a life risk. The authors aim to apply ML approaches for the early prediction of the condition in order to ease a fast clinical response.
GENERAL CONSIDERATIONS
The manuscript would benefit from a complete restructuration of the sections (e.g. where is the results section?).
Authors should make sure that all abbreviations have been defined in the text. Example: Definitions for PPG and ECG have not been found.
Numerous auto-citations have been detected (11% of the total). I encourage the authors to consider if all the auto-citations are justified.
It is not very clear if some of the figures come from other references or are original. For figures from other references, complete reference (not just in the text, but also in the caption) is needed. It is important to make sure that no copyrights are violated.
It is of main importance that the authors highlight the relevance of the specific output of their research and how it generates novelty/originality in comparison with other approaches.
COMMENTS PER SECTION
INTRODUCTION + Section 2
in the introduction, the authors clearly highlight the importance of the research, explain the causes of IDH and risk factors, mention usual actions taken against IDH and current IDH prediction approaches, both involving or not machine learning (ML) algorithms. In section 2 the authors describe the main causes of IDH in more detail, as well as the actions taken in response to (or to prevent) IDH.
- UFR is defined as ultrafiltration rate and UF as ultrafiltration. However, afterwards “UF rate” instead of “UFR” is mentioned. Example: line 50.
- Introduction and Table 1 would benefit from the addition of more related recent publications, such as (all from 2021):
- DOI: 10.2215/CJN.09280620
- DOI: 10.1109/ECBIOS51820.2021.9510559
- DOI: 10.1109/ECBIOS51820.2021.9510749
- DOI: 10.1097/MNH.0000000000000738
Table 1
- Method heading should be, for instance, “references”.
- Instead of the “Data used” heading, “data source” could be used.
MATERIALS AND METHODS + DISCUSSION + CONCLUSIONS
- I encourage the authors to consider if sections 3.1 and 3.2 should be placed in another section (introduction/discussion)
-Line 277: “All patients were 30 years of age or older” – how much older? Please specify the age range.
-Table 3, 4: I would suggest dividing Hypotension and Normal sections.
Where is the results section? Part of the materials and methods section should be presented as results and/or results and discussion section. The same for the results presented in the discussion section.
The conclusions would benefit from a clearer highlight on the output and performance obtained by the method presented during the manuscript, as well as a clear statement on what the presented approach is better/novel in comparison with others.
Author Response
Reviewer #2
SUMMARY
This work aims to predict intradialytic hypotension (IDH) using a deep learning approach fed mainly with risk-related parameters and continuous monitoring of heart rate. The authors highlight the importance of a fast and effective prediction of IDH, which affects a 15-20 % of patients undergoing dialysis and clearly represents a life risk. The authors aim to apply ML approaches for the early prediction of the condition in order to ease a fast clinical response.
GENERAL CONSIDERATIONS
The manuscript would benefit from a complete restructuration of the sections (e.g. where is the results section?).
Ans) As per your advice, we have reorganized the sections throughout the paper. Also, the Results section is located on page 7, line 246.
Authors should make sure that all abbreviations have been defined in the text. Example: Definitions for PPG and ECG have not been found. 저자는 모든 약어가 텍스트에 정의되어 있는지 확인해야 합니다. 예: PPG 및 ECG에 대한 정의를 찾을 수 없습니다.
Ans) PPG is an acronym for photoplethysmogram. On page 2, line 74, the full name of the abbreviation has been added. ECG is also an abbreviation for Electrocardiogram. On page 2, line 78, the full name of the abbreviation was added.
Numerous auto-citations have been detected (11% of the total). I encourage the authors to consider if all the auto-citations are justified.
Ans) Automatic citations have been fixed.
It is not very clear if some of the figures come from other references or are original. For figures from other references, complete reference (not just in the text, but also in the caption) is needed. It is important to make sure that no copyrights are violated.
Ans) As per your advice, references are clearly marked in each figure and description. For Figure 1, the original figure from Ref. [14] was utilized. Reference [14] was also added to the caption of Figure 1. For Figure 2, the original figure from Ref. [34] was utilized. Reference [34] was also added to the caption of Figure 2. And in Figure 5, the concept of the figures in Reference [40] was utilized. Reference [40] was also added to the caption of Figure 5.
For reference, the figures below are original figures from References [14], [34], and [40].
The original figure in Reference [14] (left) and the revised figure of Figure 1 in this paper (right)
The original figure from Reference [34] (top) and the revised figure of Figure 2 in this paper (below)
Original figure from Reference [40] (top) and modified figure of Figure 5 in this paper (bottom)
It is of main importance that the authors highlight the relevance of the specific output of their research and how it generates novelty/originality in comparison with other approaches.
Ans) As per your advice, the following has been added to Conclusion, line 498.
------------------------------------------------------------------------------------------------------------
Our approach avoids using only instantaneous data that may have a low correlation with a given IDH in a clinical setting, as in other studies. Specifically, it is based on the inference that changes in blood pressure lead to changes in the macroscopic slope of the pulse, and we found that an abrupt abnormal blood pressure reaction such as IDH results in features attributed to body homeostasis, such as symptom points.
COMMENTS PER SECTION
INTRODUCTION + Section 2
in the introduction, the authors clearly highlight the importance of the research, explain the causes of IDH and risk factors, mention usual actions taken against IDH and current IDH prediction approaches, both involving or not machine learning (ML) algorithms. In section 2 the authors describe the main causes of IDH in more detail, as well as the actions taken in response to (or to prevent) IDH.
- UFR is defined as ultrafiltration rate and UF as ultrafiltration. However, afterwards “UF rate” instead of “UFR” is mentioned. Example: line 50.
Ans) That's right, the abbreviation should be used. For the entire article, the term UF rate has been changed to UFR. thank you
- Introduction and Table 1 would benefit from the addition of more related recent publications, such as (all from 2021):
- DOI: 10.2215/CJN.09280620
- DOI: 10.1109/ECBIOS51820.2021.9510559
- DOI: 10.1109/ECBIOS51820.2021.9510749
- DOI: 10.1097/MNH.0000000000000738
Ans) As per your advice, the latest publications that you have pointed to are described below on page 2, line 93, and have also been added to Table 1. However, for the 4th paper, I could not obtain a paper because I did not have a journal account. As a result, it was excluded.
-------------------------------------------------------------------------------------------------------------
[27] proposed a recurrent neural network using a timestamp-bearing dataset to predict IDH. To improve the performance of IDH prediction, a new feature selection method using Long Short-Term Memory was proposed in [28]. [29] also utilized time-related differences in machine learning and DL methods for IDH early warning.
[27] Lee, H.; Yun, D.; Yoo, J.; Yoo, K.; Kim, Y.C.; Kim, D.K.; Oh, K.H.; Joo, K.W.; Kim, Y.S.; Kwak, N.; Han, S.S. Deep learning model for real-time prediction of intradialytic hypotension. Clin. J. Am. Soc. Nephrol. 2021, 16, 396-406.
[28] Hu, H.W.; Yang, J.Y.; Un, C.H.; Chen, K.Y.; Huang, C.C.; Tsaih, R.H. The New method of feature selection for intradialytic hypotension prediction using machine learning. IEEE 3rd Eurasia Conference on Biomedical Engineering, Healthcare and Sustainability (ECBIOS), Tainan, Taiwan, 28-30 May 2021.
[29] Yang, J.Y.; Hu, H.W.; Liu, C.H.; Chen, K.Y.; Un, C.H.; Huang, C.C.; Chen, C.C.; Lin, C.C.K.; Chang, H.; Lin, H.M. Differencing time series as an important feature extraction for intradialytic hypotension prediction using machine learning. IEEE 3rd Eurasia Conference on Biomedical Engineering, Healthcare and Sustainability (ECBIOS), Tainan, Taiwan, 28-30 May 2021.
Table 1
- Method heading should be, for instance, “references”.
Ans) Method terminology has been modified to references.
- Instead of the “Data used” heading, “data source” could be used. "
Ans) The “Data used” title has been changed to “data source”.
MATERIALS AND METHODS + DISCUSSION + CONCLUSIONS
- I encourage the authors to consider if sections 3.1 and 3.2 should be placed in another section (introduction/discussion)
Ans) Section 3.1 has been moved to Introduction, line 105, and Section 3.2 has been moved to discussion, line 420.
-Line 277: “All patients were 30 years of age or older” – how much older? Please specify the age range.
Ans) It has been modified as follows.
Line 209 : The age of the patient has a distribution of 1 in their 30s, 3 in their 40s, 16 in their 50s, 34 in their 60s, 25 in their 70s, and 10 in their 80s.
-Table 3, 4: I would suggest dividing Hypotension and Normal sections.
Ans) Tables 3 and 4 were divided into hypotensive and normal sections as shown below.
Where is the results section? Part of the materials and methods section should be presented as results and/or results and discussion section. The same for the results presented in the discussion section.
Ans) The Results section is on page 7, line 246. Section 3.1 (3.1. Real-time application) of Materials and Methods has been moved to the introduction, and Section 3.2 (3.2. ECG analysis of patients with IDH) has been moved to the discussion. And the change in HR slope before IDH occurrence (existing figure 5 and its description) was moved to section 4.1 of the results.
The conclusions would benefit from a clearer highlight on the output and performance obtained by the method presented during the manuscript, as well as a clear statement on what the presented approach is better/novel in comparison with others.
Ans) As per your advice, the following has been added to Conclusion, line 497.
------------------------------------------------------------------------------------------------------------
Our approach avoids the batch use of instantaneous data, which may have a low correlation with a given IDH in a clinical setting, as in other studies. Specifically, it is based on the inference that changes in blood pressure lead to changes in the macroscopic slope of the pulse, and we found that sudden abnormal blood pressure reactions such as IDH lead to characteristics due to homeostasis of the body, such as symptom points.
Thank you for your valuable review !!

Reviewer 3 Report
This is a very good and well written piece.
I have only the following minor comments.
1. Minimize the use of acronyms and put them into a table
2. Avoid short paragraphs, as for example the paragraph 2.3
3 Check the resolution of the figures.
4.Pheraphs the first two sections could be joined into one section.
5. After addressing (4) insert a clear and effective section with the purpose
Author Response
Reviewer #3
This is a very good and well written piece.
I have only the following minor comments.
- Minimize the use of acronyms and put them into a table.
Ans) As per your advice, we have minimized the use of abbreviations throughout the paper.
- Avoid short paragraphs, as for example the paragraph 2.3.
Ans) 5 page, 172 line : As your advice, that short 2.3 paragraph was included in 2.3.1.
3 Check the resolution of the figures.
Ans) The resolution of each figure was corrected (in particular, figures 2, 4, 7, and 8).
4.Pheraphs the first two sections could be joined into one section.
Ans) As per your advice, I've merged the first two sections into one. And the 4th and 5th sections were merged.
- After addressing (4) insert a clear and effective section with the purpose.
Ans) Section 3.1 of Materials and Methods (3.1. Real-time application) has been moved to introduction, line 105. In addition, the following content has been added to line 117 of the introduction.
----------------------------------------------------------------------------------------
In addition, our approach in this paper avoids the disadvantages of existing methods using instantaneous data given in clinical practice because some instantane-ous data may not macroscopically reflect the occurrence of IDH. Specifically, it is pro-posed that changes in macroscopic pulse gradient as well as instantaneous pulse changes can be effectively utilized to predict abnormal BF such as IDH. In addition, a sudden change in BF, such as IDH, was used in the proposed model by discovering characteristics related to the body's homeostasis, that is, the body's response to reduce BF changes.
Thank you for your valuable review !!

Round 2
Reviewer 1 Report
All my questions were addressed. I have no further questions.